# Architecting an Advanced Maturity Model for Business Processes in the Gig Economy: A Platform-Based Project Standardization

**Arfive Gandhi and Yudho Giri Sucahyo ***

Faculty of Computer Science, Universitas Indonesia, Depok 16424, Indonesia; arfive.gandhi@ui.ac.id
* Correspondence: yudho@cs.ui.ac.id

**Abstract:** The business continuity of the gig economy is strongly driven by the operator's ability to manage the maturity of business processes. Moreover, projects in the gig economy are risky due to the lack of monitoring and involvement of actors' profiles. When business processes become mature as the target, platform-based project results can satisfy actors' expectations. To reach targeted maturity, operators need to standardize their business processes. This standardization is actualized in a maturity model as a benchmark and guideline tool. It exposes how mature the current business processes are and the required improvements. This research aims to construct a maturity model systematically and comprehensively to encourage operators in the gig economy (as the model user) to improve the products and services delivered. This research has constructed a new maturity model for business processes using the maturity model development phases initiated by de Bruin et al. It explores the gig economy ecosystem in Indonesia. This research initiates the maturity model by collecting 48 factors in the gig economy. It continues by composing 13 determinant candidates as representations of the factors. After an empirical test involving 200 people (consisting of gig worker, client, and operator) and two iterations of mixed-method validation involving 16 experts, this research generates ten determinants classified into three dimensions: actors, platforms, and transactions. The maturity level of each determinant is measured to indicate its position toward digital business continuity.

**Keywords:** gig economy; maturity model; project management; gig worker; business process

---

## 1. Introduction

The gig economy is becoming a trend for digital-based work models in various worlds under the impact of globalization and digitalization (Chivite Cebolla et al. 2021). Unlike the conventional employment model, which has clarity of work and income, the gig economy offers work flexibility (Huang et al. 2020) by using platforms, both through websites and mobile applications. The gig economy actualizes a marketplace for the mediation of physical and digital services and tasks (Howcroft and Bergvall-Kåreborn 2019). The gig economy is a concrete example of a digital business that reflects how information technology (IT) can transform the work scheme for the entire world. The gig economy does not require formal education to participate but gives more prioritize the competence and interest aspects of gig workers (Gandhi et al. 2018a). Furthermore, gig workers are free to choose the type of gig project they want to participate in (Graham et al. 2017b).

Several studies in the last two years noted the occurrence of several problems related to the gig economy. The primary data in a gig economy platform showed an oversupply in three Southeast Asian countries and three African countries (Graham et al. 2017b). Oversupply refers to a high gap between gig workers with ongoing projects and registered gig workers without ongoing projects. This phenomenon certainly can reduce gig workers' motivation to continue their participation in the gig economy.

Gig economy requires synergy among entities in the ecosystem to sustain its growth as a digital business innovation. Competition among gig platforms is fierce and tight (Riley 2017). Moreover, many gig workers choose to sign up on various platforms to enlarge their opportunities to get gig projects. Their behavior is followed by clients, who open project bids on several platforms to obtain more qualified gig workers. This situation should motivate operators to manage business processes in the gig economy ecosystem reliably and effectively. Business processes should be more reliable, qualified, and satisfy the needs and motivations of gig workers and clients. However, there is no standardization of business processes that guides gig economy operators in evaluating current quality and planning for future improvements.

Many practical problems in the gig economy need solutions immediately. For example, the issue of platform misuse to earn money is a phenomenon (Brillian et al. 2018; Hunt 2015) that causes the use of a platform require intensive supervision. In addition, gig worker dissatisfaction occurs due to the unequal distribution of projects among gig workers. Many researchers reported that gig workers perceived high risks in gig economy ecosystems (Corujo 2017; Riley 2017; Tran and Sokas 2017), especially for problematic time management (Ahsan 2020; Graham et al. 2017b; Graham and Woodcock 2018; Lehdonvirta 2018) and the uncertainty of obtained projects and money (Graham et al. 2017a). Disputes between clients and gig workers when realizing products or services could easily occur because they do not meet physically and lack requirement explanations and monitoring (Du and Mao 2018; Tu et al. 2017). Aside from that, the platform's resilience to various security risks is also a concern, considering that cyber threats always haunt the digital business ecosystem, including platform abuse (Brillian et al. 2018; Hunt 2015). These issues indicate the many business processes that operators need to manage to produce a quality and reliable ecosystem amidst the different needs of each entity. Without standardization, operators can only rely on instinct to carry out the repair process sporadically. The basic form of standardization is a maturity model that reflects the growth process of operators in managing economic business processes.

Using the maturity model, operators have a baseline to appraise the current growth of business processes and the necessary improvements in the future. Additionally, maturity model has become a reliable and valuable management tool to guide an organization in identifying and developing the relevant capabilities (Häckel et al. 2021; Mantelaers and Zoet 2018). However, the existence of the maturity model requires systematic and rational phases to reflect the aspects that represent a maturing business process in a gig economy. Lahrmann et al. (2011) indicated the five essential characteristics of maturity models which should be covered when developing a maturity model: the maturity concept, the dimensions, the levels, the maturity principle, and the assessment approach. This research postulates two research questions (RQs) toward Lahrmann's maturity model characteristics:

- RQ.01: What are the determinants that influence the maturity of business processes in a gig economy?
- RQ.02: How are the determinants organized as the elements of the maturity model to fulfill the maturity principle?

This research aims to construct a maturity model systematically and comprehensively for the gig economy business process. It also aims to encourage operators in the gig economy (as the users of the maturity model) when improving the products and services delivered through the gig economy work scheme. The maturity model will become an instrument that provides quality information for operators in developing plans for continuous improvement. From a technical point of view, the maturity model will be interpreted as a direction for developing a more suitable platform to accommodate the maturity of business processes, such as through service personalization and a proper information security management system.

This research has several positions related to the research gaps. First, it fills the development of a business process maturity model specifically for the gig economy context. Second, it becomes an update to the initial maturity model of business processes as gener-

ated by Gandhi et al. (2019b), which has elements that have not been structured and have not carried out systematic validation. The third gap is related to solving technical issues in the gig economy from IT domain.

This article is composed as follows. Section 2 narrates the related theories and "quo vadis" of the maturity model in the gig economy context, while Section 3 exposes how this research performs the procedure following its classification. Section 4 unveils all findings in this research from Phase 1 to 4 from the design of de Bruin et al. (2005). Section 5 criticizes all the findings compared with the theories and relevant research. Finally, Sections 6–8 are composed of the conclusions, recommendations, and limitations, respectively.

## 2. Literature Study

### 2.1. Theories of a Gig Economy

In simple terms, Telles (2016) defined the gig economy as a digital, service-based, on-demand program that allows flexible work arrangements. In another case study, Gleim et al. (2019) highlighted that a gig economy is a labor market of ad hoc, short-term, freelancer, or otherwise non-permanent jobs. Working in a gig economy has emerged as growth of a sharing economy by leveraging project-based assignments and short-term client–worker relationships (Wairimu 2020). Ainsworth underlined the importance of platforms for the gig economy by stating that the gig economy refers to work practices that involve individuals as users of digital programs finding and carrying out short-term jobs (Ainsworth 2017). These two definitions are in line with Friedman's (2014) understanding that the gig economy is a capital–labor relationship through a digital program that connects the supply on the labor side with the demand for work on the consumer side to complete a small job. Additionally, Duggan et al. (2020) declared that the gig economy refers to an economic system that employs digital platforms to link single service providers or workers with clients. Considering all definitions, this research defines a gig economy as a work scheme in virtual environment that brings together people who utilize their assets, knowledge, or talents to find and work on remote projects in a short period of time based on certain criteria without permanent employee status. This research also visualizes gig economy ecosystem in Figure 1.

Heeks (2017) classified the gig economy into two main classes: physical and digital. A physical gig economy utilizes the platform to initiate project bidding and payment while the product or service will be delivered directly face to face. The digital gig economy (Kässi and Lehdonvirta (2018) called it as online gig economy) facilitates all parts of the transaction with a platform as a virtual medium or environment. Furthermore, Kässi and Lehdonvirta (2018) mentioned three major transformations that influenced the online gig economy: from local to remote, from full-time to temporally flexible, and from permanent to casual.

This research emphasizes the project management concept in the gig economy as a facilitated by operators to gather clients and gig workers in a virtual market. Both of them are brought together through a digital platform for flexible project work without a permanent work bond. The gig economy accommodates people's needs for extra income, even across national borders. By capitalizing on their cognitive skills and physical assets, gig workers compete in the project's tender process and realization. Communication between gig workers and clients is typically one-sided and delivered in written instructions on the project before the gig worker starts with the assigned task (Fest et al. 2021).

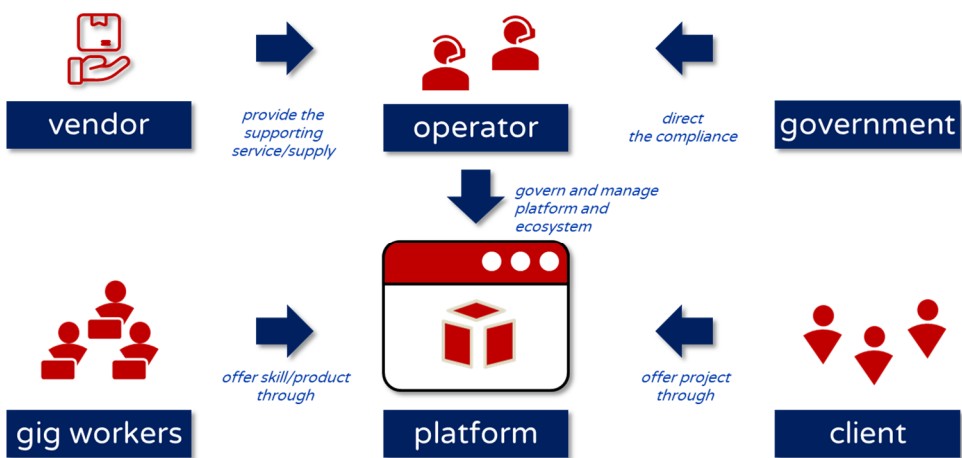

**Figure 1.** Ecosystem of gig economy.

*2.2. Theories of the Maturity Model*

Maturity model theory is a concept that describes the knowledge of standardizing the quality of business processes in an organization. This standardization is actualized in the form of several levels, which represent the worst quality to the best quality in a sequence. The aim of using this maturity model is to map the current condition of an organization in carrying out its business processes. This mapping is carried out by testing several criteria that represent a certain level of the organization's business processes. By occupying the appropriate level, the organization will be able to identify what its current strengths and weaknesses are. Furthermore, the organization will translate what is needed to achieve the higher quality represented by a higher level of maturity.

The maturity model eases the organization for formulating necessary planning and stepwise capability development (Häckel et al. 2021), since it is the assessment tool for self-appraisal or third-party evaluation (de Bruin et al. 2005; Mettler 2011). It is a trigger for the organizations to assess their current situations and provide handholds for improving their situations. In addition, Häckel et al. (2021) promoted the advantage of the maturity model through its three types: descriptive (status quo appraisal and potential target state derivation), comparative (benchmarking), and prescriptive (enabling roadmap development and suggesting measures for achieving it). By developing a maturity model suitable for business processes in the gig economy ecosystem, this study will be able to propose a maturity assessment process to gig economy operators. This maturity assessment will contribute to establish a standard for assessing the maturity and quality of the business processes of the gig economy so that operators can carry out more strategic improvement.

Maturity modeling is becoming a trend in the information systems/information technology sector after the popularity of the maturity level scheme initiated by CMMI-DEV related to software development. CMMI-DEV provides five levels that represent the maturity of an organization in managing the software development process as needed. Determining the level involves criteria grouped into several categories, including project integration, project monitoring and control, and analysis and measurement (Proença and Borbinha 2018).

*2.3. Theories of Business Processesh*

According to Milani (2019), a business process explains an organization's work, including what triggers the process, the activities carried out, the data objects used and generated, the resources that carry out the work, and the resulting outputs. This definition aligns with Stark's statement that emphasized business processes as a set of activities managed using clearly defined inputs and outputs to create business value (Stark 2020). Lohrmann and Reichert (2013) underlined business processes as a set of activities aimed at realizing business goals. Therefore, this research defines the business process as a series of organizational activities to achieve business goals.

It also highlights the vital role of business processes for organizations to achieve quality operations, both in products and services. Lohrmann and Reichert (2013) underlined that quality management is obtained through quality business processes to build service to customers appropriately. Those premises become a trigger for top management to understand how business processes are run within the organization to improve. In the gig economy context, operators need to master business processes to produce effective and agile management of the changes. Thus, understanding the gig economy business process is an absolute requirement for operators.

### 2.4. Quo Vadis Maturity Model for Gig Economy Business Processes

Gandhi et al. (2019b) published their idea to construct a maturity model for gig economy business processes (Initial MMGEBP). Many threats are faced by gig economy operators, such as information breaches, lack of gig workers' professionalism, and lack of guidance. Those threats may reduce gig operators' ability to survive. A gig economy operator should understand their current positioning in the digital business landscape. The Initial MMGEBP standardizes the gig economy's growth by considering technology acceptance, risk management, and platform development life cycles. It declares the following five levels: Initial, Defined, Standardized, Measured, and Optimized. To determine its achieved maturity level, eight staged business process areas with specific goals and detailed specific practices were set up to become instruments using specific evidence as work products (Gandhi et al. 2019b). This guidance portrays the current reflection and formulates necessary improvement toward mature and qualified business processes in a gig economy.

Compared with the Initial MMGEBP, this research explores a more holistic perspective by completing the model components with more empirical factors. It delivers more insightful and reliable determinants. This research also focuses on adjusting the measurement style by removing the process area. Previously, the Initial MMGEBP accommodated one measurement that applied for all areas. Meanwhile, this research considers measuring the maturity level for each determinant factor.

## 3. Methods

### 3.1. Research Classification

This research is included in the exploratory category because this research explores the potential factors as determinants and a validation process. In addition, factor gathering at the beginning made this study survey the research, where this study combined many preliminary studies using the primary data only. This study used a pragmatic approach in which the results obtained were consolidations and elaborations of the findings obtained using quantitative and qualitative approaches. This type of exploratory research fits within the context of maturity model construction because existing theories were still relatively immature, so they need to be explored one by one and then compiled into a compilation for further testing.

This study applied an inductive approach since the formalization process of a maturity model was performed after the identified determinants were validated by involving several experts and empirical testing with the community. The determinants were grouped after the testing process by involving experts as validators. This showed that the content of the maturity model as the answer to the research question after the exploration and validation processes.

In terms of strategy, this study applied a research survey, indicated by collecting empirical data from respondents to support the solving processes of a research question. This research actualized survey research to validate the determinants in the maturity model that involve experts. Experts conveyed their perceptions through a qualitative survey in the interview and a quantitative questionnaire. Moreover, the initial instrument's design for each determinant was also tested empirically by involving clients and gig workers.

As previously stated about qualitative and quantitative data collection, this research actualized the mixed-method approach. It was actualized in the qualitative identification of candidate factors from preliminary research followed by quantitative empirical tests involving the community. Furthermore, candidate determinants were validated by involving experts through qualitative interviews followed by a quantitative questionnaire. The validator's qualitative and quantitative perception generated more reliable and consistent data (Creswell and Creswell 2017).

*3.2. Research Phases*

This study adopted the phases of preparing a maturity model initiated by de Bruin et al. (2005). In the initiation, there were six phases, which are illustrated in Figure 2, as composed by the authors. Of the six phases (Scope, Design, Populate, Test, Deploy, and Maintain), this research only took the first four phases due to their functions. Phase 1 instructed this research to decide the maturity model's scope and basic attributes, while Phase 2 composed a proposal of maturity levels and enlisted influencing factors related with the elements of the gig economy maturity model. Phase 3 guided the process of determinant construction, representing the related factors in Phase 2. Finally, Phase 4 facilitated the testing processes in the society and the experts.

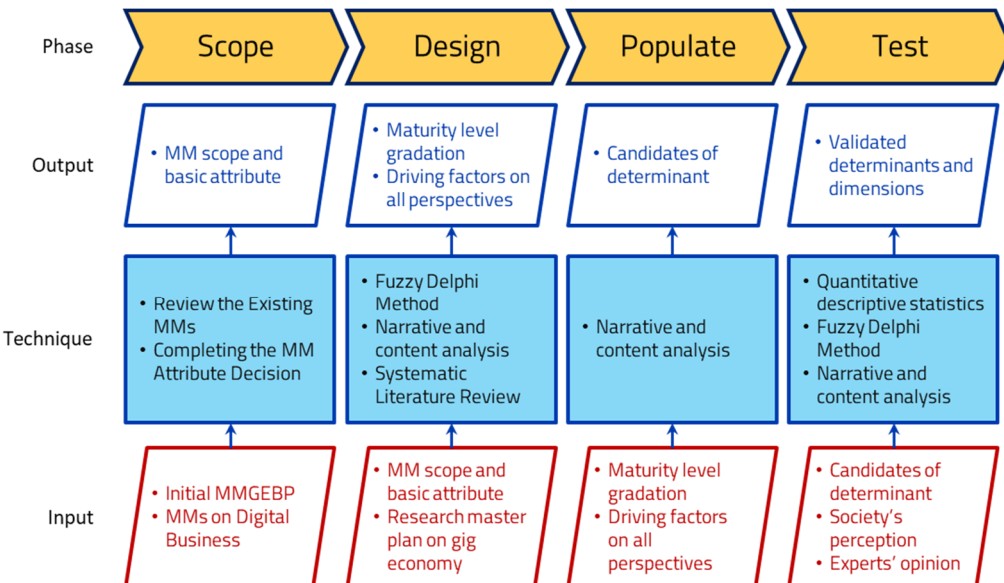

**Figure 2.** Research phases.

*3.3. Data Collection*

This research gathered three data types: secondary data for the literature review, primary data for empirical testing, and primary data for expert validation. Secondary data were derived from the published articles in the research plan in the gig economy context. It filled Phases 2 and 3 when this research enlisted the influencing factors to compose the determinant candidates. The primary data were derived from people's perceptions when appraising the research results in Phase 4. This research examined quantitatively the initial instruments of the determinant candidates with 200 people. They represented three actors in a gig economy: gig workers, clients, and operators. Their perceptions reflected whether the determinant candidates were agreed upon by society. The respondents were randomly selected by combining snowballing and convenience approaches. Data collection was carried out using online and paper-based questionnaires following the respondents' personas.

This research also examined the determinant candidates through expert judgment using the mixed-method approach. Experts represented academicians, practitioners, and

governments who appraised the elements of maturity models from academic and practical point of views. The experts were selected using a convenience approach by considering their knowledge and experience in the digital business field. Tables 1 and 2 detail their profiles as prepared by the authors.

**Table 1.** Respondents' profiles in empirical testing.

| Attribute | Criteria | Quantity | Percentage |
|---|---|---|---|
| Experience (years) | Less than 1 | 18 | 9.00 |
| | 1–2 | 48 | 24.00 |
| | 3–5 | 102 | 51.00 |
| | More than 5 | 32 | 16.00 |
| Types of Projects | Translation and writing * | 40 | 20.00 |
| | Information technology * | 58 | 29.00 |
| | Graphic, photo, and video * | 28 | 14.00 |
| | Education ** | 1 | 0.05 |
| | Ridesharing and logistics ** | 175 | 87.50 |
| Age (years) | Less than 21 | 7 | 3.50 |
| | 21 to 30 | 103 | 51.50 |
| | 31 to 40 | 61 | 30.50 |
| | More than 40 | 29 | 14.50 |
| Domicile | Jabodetabek | 88 | 44.00 |
| | Jawa Barat (non-Jabodetabek) | 55 | 27.50 |
| | Jawa Timur | 11 | 5.50 |
| | Banten (non-Jabodetabek) | 9 | 4.50 |
| | DI Yogyakarta | 8 | 4.00 |
| | Jawa Tengah | 5 | 2.50 |
| | Sumatera | 14 | 7.00 |
| | Indonesia (outside Java and Sumatera islands) | 8 | 4.00 |
| | Foreign | 2 | 1.00 |
| Role | Gig Worker | 58 | 29.00 |
| | Client | 138 | 69.00 |
| | Operator | 4 | 2.00 |

* Digital/online gig economy. ** Physical gig economy.

Regarding the sample representativeness, this research focused on both the physical and digital gig economy types (Heeks 2017) as shown in Table 1 without specifying the work field (such as transportation, software engineering, or goods delivery). In addition, the sample quality was shown in the composition of the respondents in the empirical test (i.e., clients, gig workers, and operators). Meanwhile, the composition of experts consisted of practitioners, academics, and government.

**Table 2.** Respondents' profiles in expert validation.

| Attribute | Criteria | First Iteration | Second Iteration |
|---|---|---|---|
| Experience (years) | Less than 10 | 0 | 1 |
| | 10–20 | 3 | 2 |
| | More than 20 | 5 | 5 |
| Background | Academician | 4 | 5 |
| | Practitioner | 3 | 2 |
| | Government | 1 | 1 |

## 4. Results

### 4.1. Phase 1: Scope

In the first phase, this research decided the scoping and fundamental attributes of the maturity model. This research decided on three entities as involved stakeholders: aca-

demicians, practitioners, and government. This study determined that the audience of this maturity model would be the internal operators of entity actors responsible for organizing and orchestrating business processes in the gig economy. Thus, the implementation method of maturity assessment was self-assessment with respondent representatives from operator management, operator staff, and business partners, including gig workers. Figure 3 summarizes the process of selecting the characteristics of the maturity model compiled in this study as adopted from the maturity model attributes designed by de Bruin et al. (2005).

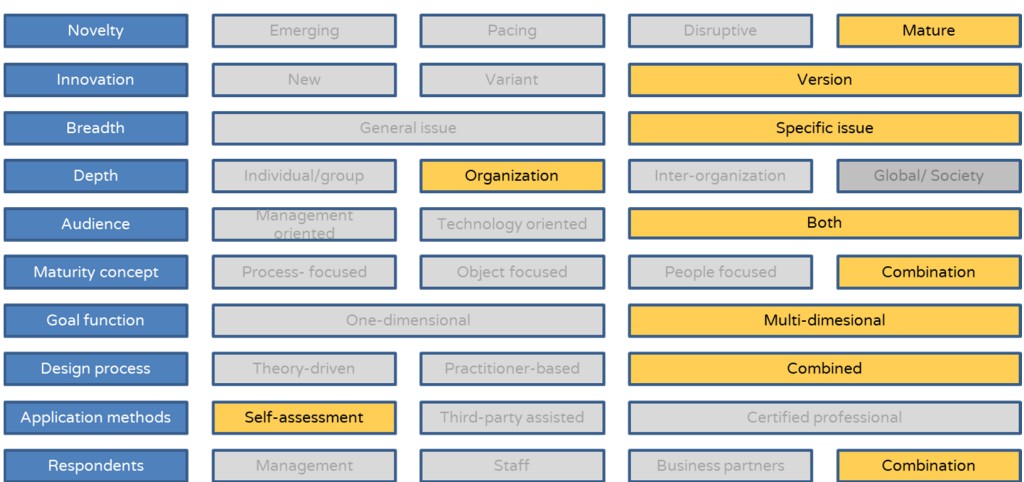

**Figure 3.** Attribute decisions in the maturity model.

### 4.2. Phase 2: Design

This study conducted a gradation design between the maturity levels and determined the domains involved when selecting the determinant candidates in the maturity model in the design phase. This maturity level gradation design was initiated by comparing the relevant and qualified maturity models in the industrial world. By making comparisons, this research can obtain the usual and rational gradations to use the maturity model in general. Then, this study created an inventory of determinant candidates by referring to related research studies in the research plan of the gig economy. The puzzle of research plan is visualized in Figure 4, while its puzzles were derived from four perspectives: gig workers, clients, operators, and project management.

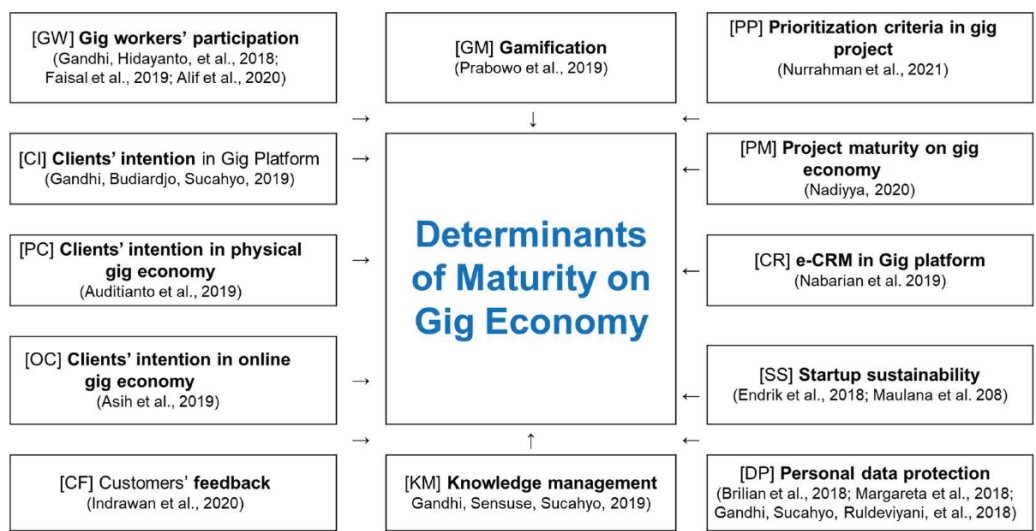

**Figure 4.** Puzzles of the research plan.

### 4.2.1. A Proposal of Maturity Level Gradation

In determining the scope, this study was initiated by formulating a proposed gradation in the maturity level of the point by comparing the established maturity models in the literature review. This research proposed five maturity levels (see Table 3) as prepared by the authors: Initial, Performed, Standardized, Quantitative Measured, and Optimized. A process would assess whether the goal was achieved or not. If achieved, the process was stated to have reached level 2. After reaching level 2, the process would be assessed for whether it had implemented standardization as required by level 3. When the criteria at level 3 were met, the process would be continued to be assessed for level 4, which emphasized the performance measurement.

**Table 3.** Proposed maturity level gradation.

| Level | Suggested Name | Definition |
|---|---|---|
| 5 | Optimized | Business process in the gig economy is optimized through continuous development |
| 4 | Quantitative Measured | Business process in a gig economy is measured to evaluate its performance and causes |
| 3 | Standardized | The business process in the gig economy has followed standardization |
| 2 | Performed | The business process in the gig economy has achieved its purpose but has not yet implemented standardization |
| 1 | Initial | The business process in the gig economy has not yet reached its purpose |

### 4.2.2. Gig Worker's Perspective

Several preliminary studies explored the factors that became determinants of the gig economy business processes from the gig workers' perspective. This accommodated the paradigm about individual processes in becoming a gig worker and how he or she maintained him or herself in the gig economy ecosystem. The acquisition process of gig workers was represented by identifying the factors that influenced people to join as gig workers. In previous research conducted by Gandhi et al. (2018a), six factors (coded as GW.01–GW.06) influenced this participation: Competence, Interest, Need for Success, Need for Independence, Economical Motivation, and Social Influence. These factors were obtained through exploratory studies relying on a qualitative approach involving gig workers in Indonesia. These factors were confirmed in two other related pieces of research: Faisal et al. (2019) and Alif et al. (2020). Moreover, Faisal et al. (2019) and Alif et al. (2020) contributed Trust, Perceived Risk, and Perceived Usefulness (coded as GW.07–GW.09).

This gig workers' perspective was in line with the findings of Maulana et al. (2018) and Endrik et al. (2018) on start-up sustainability (most gig economy operators are platform-based start-ups). They found that one of the keys to the sustainability of start-ups is human resources. In the context of the gig economy, human resources refer to the gig workers (coded as SS.02). This research argues that business processes' maturity will increase when the platform as a sustainable product (coded as SS.01) can achieve the determinants from the gig worker's perspective.

To maintain the gig workers' motivation, the operators apply a gamification system. The gamification concept refers to the adoption of elements usually found in games into a non-game ecosystem. Gamification implementation in the gig economy includes points systems, level classification, periodical challenges, and motivational messages. Previous research conducted by Prabowo et al. (2019) claimed that gamification strongly influenced the gig worker's intention and work performance. Moreover, they identified two types of regulation—internal and external—as the triggers that generated gamification. This research involved internal and external regulations (coded as GM.01 and GM.02) as sources

to construct the elements of the maturity model due to their significant importance for encouraging mature work performance.

Gandhi et al. (2019a) also found that knowledge management is a crucial issue. On the one hand, gig workers are competing. However, operators need to circulate knowledge among gig workers to improve their knowledge and skills. The gamification system is an alternative that can be adapted to facilitate sharing knowledge processes among gig workers with reward points for the contribution of that knowledge. It is hoped that workers will continue to share their knowledge and, at the same time, gain an appreciation for the contribution of this knowledge (Gandhi et al. 2019a). Thus, the chances of gig workers holding out are even more significant. Therefore, this research adapted knowledge management systems (coded as KM.01) as a source when composing the maturity model for the gig economy business process.

Additionally, this study highlights the phenomenon of personal data protection involving gig workers. Usually, employees in corporate entities have access to specific information by using applications. However, gig workers without an employment status have access to personal data belonging to customers when working on projects, such as information on clients' names, paid money, pick-up locations, and destination locations in the context of ridesharing. This phenomenon was captured by Brillian et al. (2018). Brillian et al. experimented with many gig workers in ridesharing regarding application abuse behavior. They unveiled those sanctions perceived (coded as DP.03) by gig workers and its technical countermeasure drove (coded as DP.04) whether he or she performed application abuse, such as a fake transaction. Both were picked as sources to elements in the maturity model for gig economy business processes.

### 4.2.3. Client's Perspective

From the client's perspective, there are many related preliminary studies. This preliminary research concentrated on exploring the factors that influence the choice to use the gig economy platform. Identifying these factors is necessary to determine the determinants contributing to the business's maturity in the economy from the client's perspective. This study argues that when the client's intention to use the platform increases, this represents a more mature gig economy business process. These factors were obtained based on four preliminary research examples.

The first initiation was released by Gandhi et al. (2019b) by composing the first version of the maturity model for business processes in the gig economy. When composing, they attempted to gather factors influencing clients to use the gig economy without distinguishing which were the physical or digital ones. They relied on a literature review of 10 articles. Among them, this initiative noticed 11 differentiating factors (coded as CI.01–CI.11) that indicated the maturity level or degree: Attitude, Perceived Ease of Use, Perceived Usefulness, Trust, Innovativeness or Novelty, Satisfaction, Perceived Value, Price, Reliability, Security or Privacy, and Quality.

Auditianto et al. (2019) focused on exploring the factors that influence gig economy platform usage from a client's perspective in the physical gig economy. Several factors were suspected of affecting people's decision making: Perceived Platform Quality, Trust, Social Influence, Perceived Risk, Economic Benefits, and Hedonic Motivation. After quantitative hypotheses testing, they discovered that Perceived Quality, Trust, Hedonic Motivation, and Economic Benefits contributed to the clients' intentions (coded as PC.01–PC.04) (Auditianto et al. 2019).

On the other hand, Asih et al. (2019) explored the factors affecting intensity, focusing on the digital/online gig economy. Initially, the factors that were suspected of having influence were Trust, Perceived Risk, Perceived Ease of Use, Perceived Usefulness, and Social Influence. After tracing people's perceptions with 113 respondents, Asih et al. (2019) tested these factors through the eight hypotheses. It turned out that only Trust, Perceived Usefulness, Social Influence, and Perceived Risk had been proven to affect people's intention to use the digital/online gig economy platform (coded as OC.01–OC.04).

If Brillian et al. focused on personal data protection from a gig worker's perspective, Margareta et al. unveiled the client's perspective about the same issue. Margareta et al. argued that personal data protection is a crucial issue in the gig economy since customers' personal data will be circulated to the gig workers, who are outsourcing without fixed employment (Margareta et al. 2018), as they are potential players to leak the personal data. Therefore, Margareta et al. (2018) examined whether the following factors influenced clients' perceptions about personal data protection: privacy violation experience, trusting belief, privacy concern, and risk belief. Among them, trusting belief (DP.02) was the only accepted factor and was captured as the candidate of the elements in the maturity model for the gig economy business process.

### 4.2.4. Operator's Perspective

In general, some issues need operators' attention in maintaining the sustainability of gig economy platforms through policies and their derivates about privacy (Gandhi et al. 2018b). On another side, operators should review and follow up on users' aspirations to build good relationships through e-CRM implementation. Considering the many various submitted aspirations, gig economy operators can adopt the concept of sentiment analysis on social media and platform marketplaces (Nabarian et al. 2019).

This research also involved the findings of Indrawan et al. (2020) about sentiment analysis on gig economy platforms in PlayStore with three applications: Gojek, Sampingan, and RuangGuru. Specific categories were discovered through a classification process using the Support Vector Machine (SVM), Multinomial Naïve Bayes, Complement Naïve Bayes classifier, and the Binary Relevance, Classifier Chain, and Label Power Sets methods. The categories were Problem Discovery, Praise, Feature Requests, Information Seeking, and Others (Indrawan et al. 2020).

This research covers e-CRM and user sentiment analysis as two components to be proposed as elements of the maturity model for the gig economy business process. They were labeled (CR) and (US) with one factor: (CR.01) e-CRM features and (US.01) user feedback mining, respectively. This research argued that their existence could contribute positively to a more mature business process in the gig economy ecosystem, especially the operator's ability to engage its users.

### 4.2.5. Project Management's Perspective

The gig economy performs tasks assigned by the client to a gig worker in a project. The term project in this context has a wide variety of factors, such as project outcomes, nominal costs, processing time, and required specifications. This study highlights that project maturity also affects the maturity of business processes in a gig economy. Both are strongly associated, since more mature business processes have a significant opportunity to produce more mature project work. Nadiyya (2020) measured the project maturity levels of several gig projects in Indonesia. They found that project maturity in the gig context was influenced by these factors (coded as PM.01–PM.05): time, budget, requirement, personnel, and client engagement (Nadiyya 2020).

The concept of project management was also explored by Nurrahman et al. (2021) regarding the prioritization issues in software development. Their research specifically addressed software development projects carried out in the economic gig ecosystem. Additionally, software development has become a sector or field in the gig economy with promising and significant growth. Moreover, software projects allow for remote implementation without a physical meeting between the gig worker (the developer) and the client.

Nurrahman et al. (2021) successfully identified several factors and criteria that prioritize gig economy-based software projects. This research argues that the factors and criteria they found can also apply in the context of other project areas, such as design and animation. However, projects with less complexity (such as ridesharing and food delivery) can involve fewer factors and criteria, such as time and costs. However, this

research criticizes whether those factors and criteria found by Nurrahman et al. (2021) were sufficient to accommodate the general gig economy context. Thus, this research leveraged these components from Nurrahman et al. as prioritization criteria in gig projects (coded as PP.01–PP.07): time, cost, requirements or specifications, project size or complexity, personnel, documentation, and user engagement (Nurrahman et al. 2021).

### 4.3. Phase 3: Populate

After carrying out related research relevant to the maturity model, this study recapitulated these factors in Figure 5 as composed by the authors. Several factors have explicit terminology and implicit context to be combined and generate 13 determinant candidates (see Table 4), which are thought to be indications of an increasingly mature gig economy business process. As one example, the CFM.01 determinant candidate was obtained as an accumulation of factors related to the profiles of individuals who became performance workers in terms of two motivational causative factors (Gandhi et al. 2018a): the work context of workers as human resources in start-up sustainability and the priority in the shown project. This study had the premise that CFM.01–CFM.13 was required so the business processes in the gig economy would mature.

**Table 4.** Determinants for the first iteration.

| Determinant | Sources | Purpose |
|---|---|---|
| (CFM.01) Gig Worker's Profiling | GW.01, GW.02, SS.02, PP.05 | The gig application's ability to manage gig worker profiles based on demographics, interests, skills, knowledge, and experience |
| (CFM.02) Clients' Profiling | CI.01 | The gig application's ability to manage client profiles based on special interests or needs |
| (CFM.03] Clients' Trust | OC.01, PC.02, DP.01, CI.04 | The gig application's ability to maintain client or customer trust |
| (CFM.04) Stakeholders' Satisfaction | PP.07, CI.06, GW.03 | The gig application's ability to accommodate and satisfy the needs of operators, gig workers, and clients |
| (CFM.05) Operator's Future Readiness | SS.01, CI.05 | The gig operator's ability to develop future applications through business innovation and information technology |
| (CFM.06) Platform Quality | PC.01, CI.11, CI.09, KM.01, CR.01 | The gig application's ability to meet the needs and stages of the process from the perspective of the operator, gig worker, and client or customer |
| (CFM.07) Platform Usability | CI.03, OC.02, CI.02 | The gig application's ability to be used by operators, gig workers, and clients or customers according to their functions |
| (CFM.08) Social Media Engagement | GW.06, OC.03, PC.03 | The gig operator's ability to utilize social media to influence public interest in using applications and to meet external information needs in managing applications |
| (CFM.09) Product or Service Specifications | PP.03, PP.04, PP.06, CI.07 | The gig application's ability to accommodate clarity and conformity to order or project specifications |
| (CFM.10) Project's Time Management | PP.01, GW.04 | The gig application's ability to accommodate time agreement for order or project execution time as well as the monitoring and control processes |
| (CFM.11) Project's Game Rules | GM.01, GM.02, DP.02 | The gig operator's ability to compile, implement, and update the rules of the game for the transacting parties |
| (CFM.12) Project's Economic Benefit | PP.02, GW.05, PC.04, CI.08 | Financial benefits obtained by each related party to conduct transactions in the gig application |
| (CFM.13) Risk Management | CI.10, DP.03, OC.04 | The gig operator's ability to identify and act on risks that can occur and affect the gig economy business process |

| [GW] Gig workers' participation | [OC] Clients' intention in OGE | [CI] Clients' intention in Gig Platform |
|---|---|---|
| [GW.01] Competence<br>[GW.02] Interest<br>[GW.03] Need for success<br>[GW.04] Need for independence<br>[GW.05] Economic motivation<br>[GW.06] Social influence<br>[GW.07] Trust<br>[GW.08] Perceived Risk<br>[GW.09] Perceived Usefulness | [OC.01] Trust<br>[OC.02] Perceived usefulness<br>[OC.03] Social influence<br>[OC.04] Perceived risk<br><br>**[PC] Clients' intention in PGE**<br>[PC.01] Perceived quality<br>[PC.02] Trust<br>[PC.03] Hedonic motivation<br>[PC.04] Economic benefits | [CI.01] Attitude<br>[CI.02] Perceived ease of use<br>[CI.03] Perceived usefulness<br>[CI.04] Trust<br>[CI.05] Innovativeness/ Novelty<br>[CI.06] Satisfaction<br>[CI.07] Perceived Value<br>[CI.08] Price<br>[CI.09] Reliability<br>[CI.10] Security and privacy<br>[CI.11] Quality |
| **[DP] Personal data protection**<br>[DP.01] Policies and derivates<br>[DP.02] Trust<br>[DP.03] Perceived sanction<br>[DP.04] Technical countermeasure<br><br>**[GM] Gamification**<br>[GM.01] External regulation<br>[GM.02] Internal regulation | **[PM] Project maturity**<br>[PM.01] time,<br>[PM.02] budget,<br>[PM.03] requirement,<br>[PM.04] personnel,<br>[PM.05] client engagement<br><br>**[KM] Knowledge management**<br>[KM.01] Knowledge management system | **[PP] Prioritization criteria in Gig Project**<br>[PP.01] time,<br>[PP.02] cost,<br>[PP.03] requirements/specification,<br>[PP.04] project size/complexity,<br>[PP.05] personnel,<br>[PP.06] documentation,<br>[PP.07] user engagement, |
| **[CR] e-CRM in Gig platform**<br>[CR.01] e-CRM features | **[US] User Sentiment in Gig platform**<br>[US.01] User sentiment features | **[SS] Startup sustainability**<br>[SS.01] Sustainable product<br>[SS.02] Human resources' capacity |

**Figure 5.** List of influencing factors.

*4.4. Phase 4: Test*

4.4.1. Empirical Testing

This research employed empirical testing by recruiting 200 people as respondents. They represented gig workers, clients, and operators whose perceptions were gathered from 30 October 2020 to 19 January 2021. The aim was to examine the feasibility of criteria from influencing factors as an initial instrument when finalizing the maturity model. Additionally, the related research's influencing factors had a limited scope, such as platform abuse applying to ridesharing only (Brillian et al. 2018).

The empirical testing included 47 criteria (see Appendix A) in line with 13 determinant candidates. The questionnaire on the empirical testing performed a model of calculating the level of disagreement with a five-point Likert scale. The questionnaire recap was processed using descriptive statistical techniques to measure the level of acceptance of indicators as instruments in the maturity model. Their statistical measurement using Cronbach's alpha indicated a valid result (see Table A1, Appendix A) such that all criteria could be adapted in the initial instrument for maturity level assessment.

4.4.2. Validating the Model Elements (First Iteration)

In the first iteration, this study involved eight validators with different backgrounds. The representatives were academicians, practitioners, and government workers who participated in November to December 2020. The data collection process in this first iteration was carried out using qualitative interview techniques and by filling out a quantitative questionnaire. These two techniques were combined to obtain more reliable, in-depth, and verifiable information for consistency. The quantitative questionnaire accommodated validators to express the level of their agreement with these issues:

- How much they agreed (on a scale of 0–10) that each determinant candidate indicated the maturity of the gig economy business process?
- How much they agreed (on a scale of 0–10) that each factor candidate was placed on a particular dimension?

The results of this quantitative scoring were then processed using the Fuzzy Delphi Method to convert every initial score into a specific range of values. Tables A2 and A3 (Appendix B) expose the conversion range for the scoring. By considering two determinants with a score d $\geq$ 0.2 out of 13 existing candidates, this research decided to overhaul the composition which would be evaluated in the second iteration. The composition was also reconstructed because the determinant with a score d $\geq$ 0.2 was the gig worker's profile. Previous studies have produced many points that underlie the importance of gig workers' involvement as a driver of the gig economy. If it were eliminated, the resulting model

would have a theoretical gap. Additionally, the validator who gave a low score was only one person out of eight people. This indicates that the factors with scores d $\geq 0.2$ had the potential to be accepted by reconstructing the composition with some adjustments.

4.4.3. Validating the Model Elements (Second Iteration)

There were 11 proposed determinants based on reconstruction from the first iteration's feedback in the second validation. They adapted a pattern for their terminology: "managed" and object. This was aimed at stating the minimal achievements that the operator should meet when managing the business process in the gig economy ecosystem. Table 5 unveils the 11 new determinants with their sources, related determinants in the first iteration, and purpose. Eight validators became participants in this second iteration with a mixed approach. Their perceptions were explored through qualitative interviews and a quantitative level of agreement. The technique of collecting and processing data in the second interaction used almost precisely the same style as the first iteration. The significant difference was the five-point scales in the quantitative measurement.

**Table 5.** Determinants for the second iteration.

| Determinant | Sources | Previous Determinant | Purpose |
|---|---|---|---|
| (CFM.14) Managed Gig Workers | GW.01, GW.02, GW.03, SS.02, PP.05, GM.01, GM.02, PM.04 | CFM.01, CFM.04, CFM.11 | Optimizing the gig workers' participation and empowerment to achieve their competency, responsibility, and performance |
| (CFM.15) Managed Clients | CI.01, OC.01, PC.02, DP.01, CI.04, CI.06, PP.07, PM.05 | CFM.02, CFM.03, CFM.04 | Optimizing clients' participation and experience (people who use services or products provided by gig workers) |
| (CFM.16) Managed Operator | PP.07, PS.08, SS.01, CI.05 | CFM.04, CFM.05 | Optimizing the operator's ability to organize actors, platforms, and orders in the gig economy ecosystem |
| (CFM.17) Managed Vendors | PP.07, PP.04 | CFM.04 | Manage suppliers of goods and transaction support services (such as merchants and payment channels) based on the agreement |
| (CFM.18) Managed Platform Quality | PC.01, CI.11, CI.09, KM.01, CR.01 | CFM.06 | Design, measure, and refine the platform's capabilities to meet user needs |
| (CFM.19) Managed Platform Usability | CI.03, OC.02, CI.02 | CFM.07 | Designing, measuring, and improving capabilities and usability of the platform related with effectiveness, efficiency, ease of learning, and fault tolerance |
| (CFM.20) Managed Social Media | GW.06, OC.03, PC.03 | CFM.08 | Make use of the media to publish and receive complaints from the user |
| (CFM.21) Managed Specifications | PM.03, PP.03, PP.04 | CFM.09 | Gig application capability to accommodate clarity and conformity to custom specifications |
| (CFM.22) Managed Times | PM.01, PP.01, GW.04 | CFM.10 | Plan, agree, monitor, and evaluate the processing times of orders |
| (CFM.23) Managed Benefits | PM.02, PP.02, GW.05, PC.04, CI.07, CI.08 | CFM.12 | Accommodate the fulfillment of benefits obtained by each actor for carrying out orders |
| (CFM.24) Managed Risks | CI.10, DP.02, DP.03, OC.04 | CFM.13 | Identify, assess, and manage risks associated with executing orders |

This research found that 10 determinants should be managed to meet the minimum threshold (see Tables A4 and A5, Appendix B). Only one determinant got the validators' disapproval: CFM.20 (d value = 0.261). The validators argued that its scope was not synchronized with other factors. Thus, social media management would be covered as a task that the operator had to manage. In this second iteration, the eight validators also exposed their opinions and agreement rates for the proposed dimensional grouping. They

declared agreement with a specific note: the third group was appropriate, but its label name should be changed from Order due to its ambiguity (between sort and instruction). Several validators suggested Transaction as a new label name. Thus, this second iteration produced 10 determinants as primary elements of the maturity model for the gig economy business process: Managed Gig Workers, Managed Clients, Managed Operator, Managed Vendors, Managed Platform Quality, Managed Platform Usability, Managed Specifications, Managed Times, Managed Benefits, and Managed Risks. They were grouped into three dimensions: Actor, Platform, and Transaction. Figure 6 visualizes the model components of the second iteration.

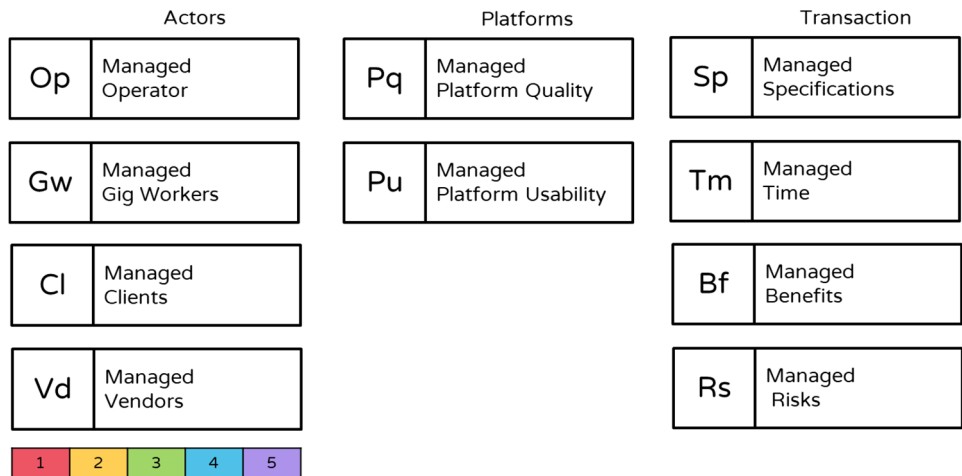

**Figure 6.** Maturity model elements.

## 5. Discussion

### 5.1. Theoretical Implications

This research produced a maturity model construction for economic business processes by following phases, as de Bruin et al. (2005) declared, albeit only in four out of six phases. The phases initiated by de Bruin et al. (2005) benefitted this research by identifying factors from various perspectives to be more holistic. This is indicated by the suitability of the maturity model with the puzzle of the research plan.

Compared with the pilot maturity model at the Initial MMGEBP, this research systematically found a more reliable determinant. If the Initial MMGEBP covered the gig worker, client, and project management perspectives (Gandhi et al. 2019b), this research completed them with an operator perspective. The validation process in the Initial MMGEBP was only performed once by involving two experts who gave qualitative feedback for (1) the proposed influencing factors, (2) the classification of the process area, and (3) the allocation of the maturity level. This indicated that the depth of the validation process was still insufficient while the number of involved experts was still small and not representative. Nevertheless, this research corrected these deficiencies by involving more experts who participated in the more detailed feedback-gathering process. Those improvements came from the usage of more structured validation techniques. This indicates that the methods and data processing techniques in developing maturity models have a significant role.

This research also applied different Likert scales between the first and second interactions in the validation process. The first iteration involved 10 Likert points, while the second iteration utilized 5 points only. This difference occurred due to feedback from the experts in the first iteration, who expressed that a 10-point scale was too rigid. Another difference lies in the change of measurement style between the first interaction and the second one. In the first iteration, each expert scored the agreement level to allocate a determinant into specific dimensions. In the second iteration, the experts scored the list of determinants contained in a dimension. It was also based on experts' feedback, as they stated that the

number of questions was too large and confusing. The learning lesson needs to consider the experts' feedback to adjust the measurement style toward their understanding.

Unfortunately, this research had not achieved the theoretical discussion Lasrado et al. (2017) issued about deciding the quantitative method of design and assessment. They criticized the variety of quantitative methods that influenced the measurement style and thinking paradigm. This research should refer them to the proper quantitative method since maturity model development is complicated and requires feasibility studies when determining its usage protocol.

*5.2. Practical Implications*

In practical implications, this research has delivered the bridge for evaluation tools for digital business to measure the growth positioning and operator's ability to manage business processes in a gig economy. This maturity model will provide the basis for operators to develop a viable business strategy. Furthermore, operators have a reference to develop platforms to satisfy each entity's needs following the business processes in the gig economy ecosystem, such as personal data protection and gamification.

## 6. Conclusions

This research has resulted in constructing a maturity model for business processes in the economy by applying the development phases initiated by de Bruin et al. (2005). To solve the problems in the background (such as risky projects and a lack of monitoring), this research identified the factors that should be accommodated using several perspectives: the gig worker, client, operator, and project management perspectives. These perspectives gathered 48 factors that were simplified into 13 determinant candidates following their context similarities. After performing the empirical tests on the society and two iterations involving experts, this study succeeded in generating 10 determinants used as measuring variables to assess how mature business processes function in the gig economy: Managed Gig Workers, Managed Clients, Managed Operator, Managed Vendors, Managed Platform Quality, Managed Platform Usability, Managed Specifications, Managed Times, Managed Benefits, and Managed Risks. Their discovery represented that RQ.01 was solved.

The experts' perceptions also grouped the 10 determinants into 3 dimensions: Actor, Platform, and Transaction. Each determinant was measured using five maturity levels: Initial, Performed, Standardized, Quantitative Measured, and Optimized. These 10 factors, 3 dimensions, and 5 maturity levels reflected the implementation of the maturity model development phase up to the fourth phase (Test) and four characteristics of the maturity model initiated by Lahrmann et al.: maturity concept, dimensions, levels, and maturity principle. Therefore, RQ.02 was solved systematically.

## 7. Future Insight

This research completed four of the six phases of maturity development initiated by de Bruin et al. (2005). The following phases (Deploy and Maintain) become the target of future research. This research pursued the proof of concept when measuring the maturity level in real case studies as the next plan. This will prove whether the construction of the maturity model can meet the suitability between theory and practice. This research also noticed that assessment instruments were not completed yet with measurement procedures. Thus, the procedures and protocol for technical assessments become the research agenda for the future. There is also the aim to fulfill Lahrmann's claimed five important characteristics of maturity models: the maturity concept, the dimensions, the levels, the maturity principle, and the assessment approach (Lahrmann et al. 2011). This research succeeded in achieving four of these, while the assessment approach did not exist.

## 8. Limitations

This research has several limitations. First, the research objects covered Indonesia only, including 200 people in the empirical test and 16 people in the expert validation. Second,

the results of element construction need to be followed up with the assessment schemes and indicators while measuring the level of maturity. Finally, this study was limited to only the gig economy facilitated by platforms, excluding social media platforms that might be used as a medium for gig transactions.

**Author Contributions:** Conceptualization, A.G. and Y.G.S.; Formal analysis, A.G. and Y.G.S.; Funding acquisition, Y.G.S.; Investigation, A.G.; Methodology, A.G.; Project administration, Y.G.S.; Resources, Y.G.S.; Supervision, Y.G.S.; Validation, Y.G.S.; Visualization, A.G.; Writing—original draft, A.G.; Writing—review & editing, Y.G.S. All authors have read and agreed to the published version of the manuscript.

**Funding:** This research was funded by PUTI Q2 grant "Digitalization on Gig Worker: A Manifestation of Digital Economy for Indonesia in Industry 4.0 Era" (NKB-1477/UN2.RST/HKP.05.00/2020). We would express our gratitude to the Faculty of Computer Science and Directorate of Research and Community Engagement at the Universitas Indonesia.

**Conflicts of Interest:** The authors declare no conflict of interest.

## Appendix A

*Appendix A.1. List of Criteria in Empirical Testing*

- (CFM.01.01) The platform verifies the gig worker data.
- (CFM.01.02) The platform manages the gig worker profiles by origin region.
- (CFM.01.03) The platform manages the profiles of gig workers based on their interests, knowledge, and skills.
- (CFM.01.04) The platform manages the work experience of gig workers as a portfolio that demonstrates quality.
- (CFM.02.01) The platform verifies the client data.
- (CFM.02.02) The platform manages client profiles based on interests.
- (CFM.02.03) The platform applies personalization to the clients' specific needs.
- (CFM.03.01) The operator maintains client trust in platform usage.
- (CFM.03.02) The operator maintains the platform's reputation and credibility for the client.
- (CFM.03.03) The involved third parties maintain the platform's reputation and credibility.
- (CFM.04.01) The platform performs stages of the process to meet gig worker satisfaction.
- (CFM.04.02) The platform runs a process stage to meet client satisfaction.
- (CFM.04.03) The platform runs a process stage to meet the operator's satisfaction.
- (CFM.05.01) The operator is actively evaluating and following up on the gig workers and clients' needs.
- (CFM.05.02) The operator has future platform development plans.
- (CFM.05.03) The operator has future business innovations.
- (CFM.05.04) The operator is prepared to improve the quality of gig workers in the future.
- (CFM.06.01) The platform performs activities reliably.
- (CFM.06.02) The platform accommodates processes according to the needs of the gig worker and client.
- (CFM.06.03) The platform meets the requirements and quality standards expected by the gig worker and client.
- (CFM.06.04) The platform provides a feature for sharing knowledge between gig workers.
- (CFM.06.05) The platform provides features to manage relationships among the operator, gig worker, and client.
- (CFM.07.01) The platform is used easily and quickly learned.
- (CFM.07.02) The platform provides benefits for gig workers and clients.
- (CFM.07.03) The platform efficiently facilitates the stages of the gig economy process.
- (CFM.08.01) The operator manages social media channels to promote gig worker services and products.

- (CFM.08.02) The operator identifies patterns of behavior, needs, and client satisfaction.
- (CFM.08.03) The operator utilizes social media for relevant processes, such as verifying data, behavior patterns, needs, and client satisfaction.
- (CFM.09.01) The demand for projects in the gig economy is clearly stated and documented.
- (CFM.09.02) The platform accommodates the monitoring process of order or project progress.
- (CFM09.03) The platform provides a checking process for project suitability and realization.
- (CFM.09.04) The platform accommodates the process of handling changes to order or project specifications.
- (CFM.09.05) The platform has project management tools to assist gig workers in handling project specifications.
- (CFM.10.01) There is an agreement on the completion of the project between the gig worker and client.
- (CFM.10.02) The platform provides reminders for gig workers to complete projects.
- (CFM.10.03) The platform accommodates the process of handling changes to the schedule for project work.
- (CFM.11.01) The operator applies a point system to gig workers based on performance.
- (CFM.11.02) The operator applies a point system to clients based on the transactions.
- (CFM.11.03) The operator updates the game rules regarding projects according to the current situation.
- (CFM.11.04) The operator imposes sanctions on both gig workers and clients if they violate the game's rules.
- (CFM.12.01) Transactions executed on the platform provide competitive prices or fees for clients.
- (CFM.12.02) Transactions generated through the platform provide financial benefits to gig workers.
- (CFM.12.03) Transactions executed on the platform provide financial benefits to the operator.
- (CFM.13.01) The operator identifies possible negative risks.
- (CFM.13.02) The operator has adequate policies, standards, and procedures to overcome the negative risks.
- (CFM.13.03) The operator is agile in following up on negative risks.
- (CFM.13.04) The operator can detect misuse of information technology.

*Appendix A.2. Empirical Testing Results*

**Table A1.** Empirical testing results.

| Criteria | CAID | Mean | Criteria | CAID | Mean | Criteria | CAID | Mean |
|---|---|---|---|---|---|---|---|---|
| CFM.01.01 | 0.956 | 4.460 | CFM.05.04 | 0.956 | 4.360 | CFM.09.05 | 0.955 | 4.210 |
| CFM.01.02 | 0.957 | 4.090 | CFM.06.01 | 0.956 | 4.385 | CFM.10.01 | 0.956 | 4.290 |
| CFM.01.03 | 0.956 | 4.185 | CFM.06.02 | 0.956 | 4.430 | CFM.10.02 | 0.955 | 4.330 |
| CFM.01.04 | 0.956 | 4.285 | CFM.06.03 | 0.956 | 4.360 | CFM.10.03 | 0.955 | 4.245 |
| CFM.02.01 | 0.956 | 4.455 | CFM.06.04 | 0.956 | 3.860 | CFM.11.01 | 0.956 | 4.365 |
| CFM.02.02 | 0.957 | 4.080 | CFM.06.05 | 0.956 | 4.365 | CFM.11.02 | 0.956 | 4.280 |
| CFM.02.03 | 0.956 | 4.340 | CFM.07.01 | 0.956 | 4.560 | CFM.11.03 | 0.956 | 4.225 |
| CFM.03.01 | 0.956 | 4.465 | CFM.07.02 | 0.956 | 4.645 | CFM.11.04 | 0.956 | 4.390 |
| CFM.03.02 | 0.956 | 4.475 | CFM.07.03 | 0.956 | 4.420 | CFM.12.01 | 0.956 | 4.370 |
| CFM.03.03 | 0.956 | 4.225 | CFM.08.01 | 0.956 | 4.325 | CFM.12.02 | 0.956 | 4.455 |
| CFM.04.01 | 0.956 | 4.225 | CFM.08.02 | 0.956 | 4.355 | CFM.12.03 | 0.956 | 4.355 |

**Table A1.** *Cont.*

| Criteria | CAID | Mean | Criteria | CAID | Mean | Criteria | CAID | Mean |
|---|---|---|---|---|---|---|---|---|
| CFM.04.02 | 0.956 | 4.435 | CFM.08.03 | 0.957 | 4.190 | CFM.13.01 | 0.955 | 4.175 |
| CFM.04.03 | 0.956 | 4.125 | CFM.09.01 | 0.956 | 4.375 | CFM.13.02 | 0.955 | 4.280 |
| CFM.05.01 | 0.955 | 4.425 | CFM.09.02 | 0.955 | 4.485 | CFM.13.03 | 0.956 | 4.210 |
| CFM.05.02 | 0.956 | 4.485 | CFM.09.03 | 0.955 | 4.270 | CFM.13.04 | 0.956 | 4.230 |
| CFM.05.03 | 0.956 | 4.450 | CFM.09.04 | 0.956 | 4.240 | | | |

## Appendix B

**Table A2.** Fuzzy Delphi scoring for determinant agreement in the first iteration by expert judgment.

| Attribute | CFM.01 | CFM.02 | CFM.03 | CFM.04 | CFM.05 | CFM.06 | CFM.07 | CFM.08 | CFM.09 | CFM.10 | CFM.11 | CFM.12 | CFM.13 |
|---|---|---|---|---|---|---|---|---|---|---|---|---|---|
| $d \leq 0.2$ | 0.261 | 0.178 | 0.094 | 0.000 | 0.094 | 0.178 | 0.178 | 0.094 | 0.000 | 0.000 | 0.000 | 0.261 | 0.000 |
| $d \leq 0.2$ Construct | 0.103 | | | | | | | | | | | | |
| % $d \leq 0.2$ | 0.00 | 87.50 | 87.50 | 100 | 87.50 | 87.50 | 87.50 | 87.50 | 100 | 100 | 100 | 0.00 | 100 |
| Exp Grp Consensus | 78.85% | | | | | | | | | | | | |

**Table A3.** Fuzzy Delphi scoring for dimension agreement in the first iteration by expert judgment.

| Attribute | CFM.01 | CFM.02 | CFM.03 | CFM.04 | CFM.05 | CFM.06 | CFM.07 | CFM.08 | CFM.09 | CFM.10 | CFM.11 | CFM.12 | CFM.13 |
|---|---|---|---|---|---|---|---|---|---|---|---|---|---|
| $d \leq 0.2$ | 0.094 | 0.094 | 0.161 | 0.263 | 0.261 | 0.000 | 0.178 | 0.161 | 0.000 | 0.000 | 0.000 | 0.161 | 0.000 |
| $d \leq 0.2$ Construct | | | | | | | 0.106 | | | | | | |
| % $d \leq 0.2$ | 87.50 | 87.50 | 75.00 | 37.50 | 0.00 | 100 | 87.50 | 75.00 | 100 | 100 | 100 | 75.00 | 100 |
| Exp Grp Consensus | | | | | | | 78.85% | | | | | | |

**Table A4.** Fuzzy Delphi scoring for determinants in the second iteration by expert judgment.

| Attribute | CFM.14 | CFM.15 | CFM.16 | CFM.17 | CFM.18 | CFM.19 | CFM.20 | CFM.21 | CFM.22 | CFM.23 | CFM.24 |
|---|---|---|---|---|---|---|---|---|---|---|---|
| Indicator $d \leq 0.2$ | 0.094 | 0.118 | 0.055 | 0.000 | 0.000 | 0.000 | 0.261 | 0.150 | 0.055 | 0.204 | 0.055 |
| $d \leq 0.2$ Construct with CFM.20 | | | | | | 0.083 | | | | | |
| $d \leq 0.2$ Construct without CFM.20 | | | | | | 0.065 | | | | | |
| % $d \leq 0.2$ | 100 | 87.50 | 100 | 100 | 100 | 100 | 12.50 | 87.50 | 87.50 | 100 | 87.50 |
| Exp Grp Consensus with CFM.20 | | | | | | 87.50% | | | | | |
| Exp Grp Consensus without CFM.20 | | | | | | 95.00% | | | | | |

**Table A5.** Fuzzy Delphi scoring for dimensions in the second iteration by expert judgment.

| Attribute | Actor | Platform | Order |
|---|---|---|---|
| $d \leq 0.2$ Construct20 | 0.000 | 0.055 | 0.164 |
| % $d \leq 0.2$ | 100 | 87.50 | 87.50 |
| Exp Grp Consensus | | 91.67% | |

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
