# Peer review of "Architecting an Advanced Maturity Model for Business Processes in the Gig Economy: A Platform-Based Project Standardization"

_economies, doi:10.3390/economies9040176_

Round 1

Reviewer 1 Report

This study constructed a maturity model for business processes on gig economy. It is an interesting topic. However, there are some flaws that need to be further improved. Specific comments are as follows.

  1. Abstract. The empirical analysis (data source) in this article was carried out in a specific region (Indonesia?). It is better to mention this in the Abstract.
  2. The Literature Study section is not thick enough, especially Theories of Gig Economy. I would advise the author(s) to better organize the Introduction and Literature review sections. Please provide more detailed information about the research aims of your paper.
  3. It seems to me the references are a bit outdated and not sufficient. To my knowledge, many studies on gig economy have been published between 2019 and 2021. I suggest author(s) to provide more latest references regarding the theme of this paper.
  4. Methods. The sample size might be a problem. In Table 8(Types of projects), it seems that the samples about the gig economy include 5 categories: Translation and writing, Information technology, Graphic, photo, and video, Education, and Ridesharing and logistics. However, the scope of gig economy business is very broad, such as accommodation, caregiving, home service, health services, legal services, retail. Please further explain the basis and reason for the classification used in this study. 
  5. No “Results” Section?  Why?  The Discussions section is not well structured and difficult to read. In addition, it contains too many abbreviations and does not provide corresponding detailed explanations. The author(s) should revise and improve the Results and Discussion sections to make them more logical and clearer.
  6. Typos. Line 12” de Bruin.” But Line 205 “de De Bruin”. Figure 3 (Puzzles of Research Master Plan) needs to be improved.
  7. It is better to add short paragraphs to discuss the theoretical and practical implications of the research, as well as the limitations and future research directions.
  8. This paper is not well organized. Please revise and make it clearer and more logical. In addition, pay attention to formatting manuscript according to the journal guidelines.

I hope these comments will be helpful.

Author Response

Dear Editor of Economies (MPDI)

Regarding the reviewer’s comments to our paper entitled “Architecting Advanced Maturity Model for Business Processes on Gig Economy: A Platform-based Project Standardization” we have addressed the comments as mentioned below.

Best regards,

Manuscript ID            :       economies-1397218

Type of manuscript :       Article

Title                               :       Architecting Advanced Maturity Model for Business Processes on Gig Economy: A Platform-based Project Standardization

Reviewer 2 Report

On the page 2 we can find the goal of the paper which says: “This study aims to construct a maturity model systematically and comprehensively for the gig economy business process”. In this context the fundamental question arises. Namely: who needs this maturity model and for which purpose it could be used. It’s also not clear where we can find the final maturity model (figure 5?) and what practical implications result from this model (section “Conclusion” says nothing about it). There are also some “technical” issues:

  • How 200 people involved in the research (page 5-6) where selected?
  • How experts (page 6) where selected?
  • Some of the figures are illegible (figure 2).

Author Response

Cover Letter (Responses to the Reviewer’s Comments)

Dear Editor of Economies (MPDI)

Regarding the reviewer’s comments to our paper entitled “Architecting Advanced Maturity Model for Business Processes on Gig Economy: A Platform-based Project Standardization” we have addressed the comments as mentioned below.

Best regards,

Reviewer 3 Report

Dear author,

I was pleased to read this article which provides promising research. Their work is interesting and addresses an emerging issue in the area of The Digital Economy. The authors study " Architecting Advanced Maturity Model for Business Processes on Gig Economy: A Platform-based Project Standardization”.

I felt confident that the authors performed careful and thorough field processing. However, I would like to point out some aspects to improve their manuscript. I explain my concerns in more detail below. I ask that the authors specifically address each of my comments in their response.

Major comments

1) In the abstract you should better clarify the objective of the research. You should also state more precisely the results and conclusions obtained in your study. On the other hand, when you mention the work of "De Bruin", you should refer to the scientific publication that includes this previous work. If this reference is to line (665), the following should be indicated: De Bruin et al. (2005).

2) In the introduction, it is recommended to better clarify the objective of the research carried out and its original contribution to the advancement of knowledge.

3) In section "2.1. Theories of Gig Economy" it is recommended that some additional definition be included to strengthen the theoretical framework. We recommend the incorporation of the following definition provided by Jorge-Vázquez (2019)*:

“Models of consumption and provision of services are usually based on exchange relationships between professionals and consumers (B2C) which, through digital platforms that do not provide the underlying service and act as intermediaries, a service is provided based on the needs demanded by the user in exchange, normally, for economic consideration” (pag.17-18)

*Jorge-Vázquez, J. (2019). La economía colaborativa en la era digital: Fundamentación teórica y alcance económico. In Economía Digital y Colaborativa: Cuestiones Económicas y Jurídicas; Náñez, S.L., Ed.; Università degli Studì Suor Orsola Benincasa. Eurytonpress: Naples, Italy, 2019. ISBN 9788896055915. [Avalaible at: https://cutt.ly/YEZXeP0]

4) The background research review is insufficient. The authors should make a greater effort to complete this review. Some recent references on the state of the art are suggested that may help to complete the review:

-Gleim, M. R., Johnson, C. M., & Lawson, S. J. (2019). Sharers and sellers: A multi-group examination of gig economy workers' perceptions. Journal of Business Research98, 142-152. https://doi.org/10.1016/j.jbusres.2019.01.041

-Chivite Cebolla, M. P., Jorge Vázquez, J., & Chivite Cebolla, C. M. (2021). Collaborative economy, a society service? Involvement with ethics and the common good. Business Ethics: A European Review. https://doi.org/10.1111/beer.12339. [Avalaible at: https://bit.ly/3zr0fJ1]

-Fest, S., Kvaløy, O., Nieken, P., & Schöttner, A. (2021). How (not) to motivate online workers: Two controlled field experiments on leadership in the gig economy. Leadership Quarterly forthcoming. https://doi.org/10.1016/j.leaqua.2021.101514

5) On methodology, its description needs to be improved. You refer to the use of surveys as a data collection instrument. However, these surveys are not described (design, scales, type of questions, etc.). The dates on which the data collection took place are also not stated.

On the other hand, the territorial unit of analysis is not expressly indicated, nor are its choice and characteristics justified. From the reading of the text, everything indicates that it could be Indonesia, but it is not specified. It is recommended that this question be clarified.

The authors indicate that phases 5 and 6 were excluded in this research, but they do not indicate which phases they are, nor do they adequately justify their exclusion from the research. Although the authors indicate that these will be future lines of research, it is suggested that this issue be clarified.

6) The article includes the following sections: “4.Discussion" and "5. Discussion and implication". In my opinion, this organisation of the content could be improved. It is recommended to separate the results from the discussion. In particular, create a results section and a discussion section. In the results section, It should provide a concise and precise description of the experimental results, their interpretation, as well as the experimental conclusions that can be drawn. In addition, it is recommended to include the results of the survey. On the other hand, in the discussion section, the authors should discuss the results and how they can be interpreted from the perspective of previous studies and of the working hypotheses. The findings and their implications should be discussed in the broadest context possible. Future research directions may also be highlighted.

Minor comments:

1) In the bibliography you should review the citation rules proposed by the journal in the template. In particular, the proposed format is not followed and the DOI is not included. Authors should "Include the digital object identifier (DOI) for all references where available".

2) It is recommended to indicate the source in the figures and tables included in the article, even if they have been prepared by the authors.

3) It is suggested to include a paragraph indicating the limitations of the study.

In summary, I thank you again for giving me this opportunity to learn from your research project and wish you all the best.

Best regards,

Author Response

(The authors gave the same response as above.)

Round 2

Reviewer 1 Report

Comments 2, 3, and 4 have not been well resolved. Please carefully revise the article and give a detailed explanation in the “response” file.

Author Response

Dear Editor of Economies (MPDI)

Regarding the reviewer’s comments to our paper entitled “Architecting Advanced Maturity Model for Business Processes on Gig Economy: A Platform-based Project Standardization” we have addressed the comments as mentioned below. 

Best regards,

Arfive Gandhi, Yudho Giri Sucahyo

Comment

Response

Location

R1.02a

The Literature Study section is not thick enough, especially Theories of Gig Economy. I would advise the author(s) to better organize the Introduction and Literature review sections.

Section 1 and 2 has been improved

·         Emphasize the existence of research goal/aim in the Introduction

·         Emphasize the research gaps in the Introduction

·         Complete the theories of gig economy with more qualified content, including its classification

·         Add the new subsection: Theories of Business Processes

Section 1 and 2

R1.02b

Please provide more detailed information about the research aims of your paper.

This research aims to construct a maturity model systematically and comprehensively for the gig economy business process. It also aims to encourage operator in gig economy (as the user of maturity model) when improving the products and services de-livered through the gig economy work scheme

Section 1 (line 90)

R1.03

It seems to me the references are a bit outdated and not sufficient. To my knowledge, many studies on gig economy have been published between 2019 and 2021. I suggest author(s) to provide more latest references regarding the theme of this paper.

Several relevant references have been updated

·         Chivite Cebolla (2021) about the existence of gig economy

·         Gleim et al. (2019) and Wairimu (2020) in gig economy definition

·         Fest (2021) in mechanism of gig economy

·         Stark (2020) and Milani (2019) in business process concept

Section 1, 2.1, and 2.3

R1.04

Methods. The sample size might be a problem. In Table 8(Types of projects), it seems that the samples about the gig economy include 5 categories: Translation and writing, Information technology, Graphic, photo, and video, Education, and Ridesharing and logistics. However, the scope of gig economy business is very broad, such as accommodation, caregiving, home service, health services, legal services, retail. Please further explain the basis and reason for the classification used in this study.

This research focused on two types of gig economy: physical and online, refer to the Heeks’ classification (2017). Both were captured in this research. Also, entities in gig economy have been represented: client, gig worker, and operator.

Section 3.3 (Data Collection, line 276 to 281)

Reviewer 2 Report

I believe the manuscript has been sufficiently improved to warrant publication in Economies.

Author Response

Dear Editor of Economies (MPDI)

Regarding the reviewer’s comments to our paper entitled “Architecting Advanced Maturity Model for Business Processes on Gig Economy: A Platform-based Project Standardization” we have addressed the comments as mentioned below. Green cells indicate the current change following the second round instructions.

Best regards,

Arfive Gandhi, Yudho Giri Sucahyo

Reviewer 3 Report

Dear authors,

Despite the effort made by the authors to improve the article, I still find several observations suggested initially that have not been answered by the authors or that their revision has been partial or incomplete. It is suggested that the authors check whether the version uploaded to the platform is correct. I express these observations below:

1. ABSTRACT: The research objective has not been better clarified. Nor have the results and conclusions drawn from the study been more precisely stated.

2. INTRODUCTION: The objective of the research has not been clarified. In my opinion, it is somewhat unclear and needs to be formulated more precisely.

3. THEORETICAL FRAMEWORK: I consider that the definitions included are insufficient to reinforce the section on "2.1. Theories of Gig Economy". It is recommended to add a more complete definition as suggested in the first report or similar.

4. METHODOLOGY: the description is still incomplete:

4.1. There is no description of the characteristics of the questionnaire used: design, scales applied, types of questions, etc.

4.2. The choice and characteristics of the territorial unit of analysis are not justified: is it Indonesia, why Indonesia, and what characteristics make Indonesia attractive for study?

4.3. The authors do not adequately justify the exclusion of phases 5 and 6.

5. DISCUSSION: Authors should discuss the results and how they can be interpreted from the perspective of previous studies.

I hope these observations will help you to improve your manuscript. Wish you all the best. Best regards,

Author Response

Dear Editor of Economies (MPDI)

Regarding the reviewer’s comments to our paper entitled “Architecting Advanced Maturity Model for Business Processes on Gig Economy: A Platform-based Project Standardization” we have addressed the comments as mentioned below. Green cells indicate the current change following the second round instructions.

Best regards,

Arfive Gandhi, Yudho Giri Sucahyo

Comment

Response

Location

R3.05

ABSTRACT: The research objective has not been better clarified. Nor have the results and conclusions drawn from the study been more precisely stated.

This research aims to construct a maturity model systematically and comprehensively to encourage operator in gig economy (as the user of maturity model) when improving the products and services delivered.

Abstract, line 9 to 12

R3.06

INTRODUCTION: The objective of the research has not been clarified. In my opinion, it is somewhat unclear and needs to be formulated more precisely.

This research aims to construct a maturity model systematically and comprehensively for the gig economy business process. It also aims to encourage operator in gig economy (as the user of maturity model) when improving the products and services de-livered through the gig economy work scheme

Section 1 (line 90)

R3.07

THEORETICAL FRAMEWORK: I consider that the definitions included are insufficient to reinforce the section on "2.1. Theories of Gig Economy". It is recommended to add a more complete definition as suggested in the first report or similar.

The revised article exposes the final definition about gig economy (after comparison among previous definitions) and Heeks’ statement about two types of gig economy (physical and online)

Section 2.1

R3.08

METHODOLOGY: There is no description of the characteristics of the questionnaire used: design, scales applied, types of questions, etc.

The questionnaire on the empirical testing performed a model of calculating the level of disagreement with a 5-point Likert scale. The questionnaire recap was processed using descriptive statistical techniques to measure the level of acceptance of indicators as instruments in the maturity model.

(In Validation Model 1st iteration) Quantitative questionnaire accommodated validators to express the level of their agreement for these issues:

•             How much agree (scale 0 to 10) each determinant candidate indicates the maturity of the gig economy business process.

•             How agree (scale 0 to 10) each factor candidate is placed on a particular dimension.

The results of this quantitative scoring are then processed using the Fuzzy Delphi Method technique, where each score would be converted into specific range value. Tables 7 and 8 (Annex B) expose the conversion range for the scoring.

(In Validation Model 2nd iteration) The technique of collecting and processing data in the second interaction uses almost precisely the same style as the first iteration. The significant difference is the five-point scales in quantitative measurement.

Section 4.4.1, 4.4.2, and 4.4.3

R3.09

METHODOLOGY: The choice and characteristics of the territorial unit of analysis are not justified: is it Indonesia, why Indonesia, and what characteristics make Indonesia attractive for study?

As told in Limitation, the research covered Indonesian only since gig economy has been promising alternative work scheme in developing countries. Hence, this research assumed that other developing countries had similar landscape.

Section 8

R3.10

METHODOLOGY: The authors do not adequately justify the exclusion of phases 5 and 6.

Phases 5 and 6 focus on making measuring tools consisting of schemas and assessment forms, while this research only focuses on exploring determinants, dimensions, and gradations as elements of the maturity model.

Section 3.2

R3.11

DISCUSSION: Authors should discuss the results and how they can be interpreted from the perspective of previous studies.

The MMGEBP as initial model has been compared to this research’s results

Section 5.1

Round 3

Reviewer 1 Report

The authors have addressed my previous comments. I have no further comments.

Reviewer 3 Report

Dear authors,

I think the authors have made a remarkable effort to improve the article. They have improved the structure and organisation of the content. The aim of the study has also been clarified. On the other hand, the review of the research background has been suitably expanded. The authors have also made an effort to improve the description of the methodology used and the limitations of their study.

Finally, in my opinion, the discussion of results can still be improved by incorporating some additional previous studies to enrich the discussion.

Wish you all the best. Best regards,

This manuscript is a resubmission of an earlier submission. The following is a list of the peer review reports and author responses from that submission.